Matters arising

# Reply to: Increase of P-wave velocity due to melt in the mantle at the Gakkel Ridge

Ivan Koulakov [1,2] ✉, Vera Schlindwein [3,4], Mingqi Liu [5], Taras Gerya [5], Andrey Jakovlev [1] & Aleksey Ivanov [2]

REPLYING TO Z. Yu & S. C. Singh Nature Communications https://doi.org/10.1038/s41467-023-36551-8 (2023)

Yu and Singh[1] argued that our seismic tomography model for the area of the Gakkel Ridge is not trustworthy and does not provide solid evidence for the proposed scenario of explosive volcanism development. They doubt the adequacy of the initial data analysis and express concerns that the SP phases might be associated with wave conversions in a sediment layer. In our response, we present several arguments on why it is highly unlikely. Yu and Singh[1] also presented the Wadati diagram with a very low value of the $Vp/Vs$ ratio. We have

Fig. 1 | Travel time analysis. a Travel times of the observed and calculated travel times of the $P$ and $S$ waves after source locations in the final 3D model versus hypocentral distances. b Wadati diagrams for the cases of experimental data (red dots) and travel times calculated in the final tomography model (blue dots). In both cases, the linear regression coefficients are equal to 1.71. c Wadati diagrams for the data constructed in two synthetic models with $Vp/Vs$ equal to 2.0 (red dots) and 1.7 (blue dots). The regression coefficients are indicated with the corresponding colors.

[1]Trofimuk Institute of Petroleum Geology and Geophysics SB RAS, Prospekt Koptyuga, 3, 630090 Novosibirsk, Russia. [2]Institute of the Earth's Crust SB RAS, Irkutsk, Russia. [3]Alfred Wegener Institute, Helmholtz Centre for Polar and Marine Research, Am Alten Hafen 26, D 27568 Bremerhaven, Germany. [4]Department of Geosciences, University of Bremen, Bremen, Germany. [5]Department of Earth Sciences, ETH Zurich, Sonneggstrasse 5, 8092 Zurich, Switzerland. ✉ e-mail: KoulakovIY@ipgg.sbras.ru

**Fig. 2 | Examples of seismic records on the sea bottom and ice floe in two sites.** The upper traces with names starting with "GKD" correspond to an earthquake recorded by four ocean bottom seismometers at a volcano at 120°E Gakkel Ridge. The other traces with names starting with "G85" demonstrate an earthquake recorded at the 85°E volcano by seismometers on ice floes. For display, the amplitudes of station GKD04 were downscaled by a factor of 5 and for stations G8530-33 by a factor of 10 compared to the other records of the same earthquake. A bandpass filter of 3–15 Hz was applied to all traces. The approximate positions of the P and S phase arrivals are marked by blue and red triangles, respectively.

demonstrated that they used an inappropriate method to construct this diagram and that the actual values of $Vp/Vs$ are reasonable. Yu and Singh[1] doubted about the validity of our interpretation and based their arguments on the effective medium theory[2]. We admit that our seismic model cannot be directly converted to petrophysical parameters due to the uncertainty of amplitude determination and should be interpreted qualitatively.

The main concern of Yu and Singh[1] is that we got the wrong input data for tomography. They suspected that during the data processing, we misidentified the $SP$ phase and picked another phase having no relation to the $S$-wave in deep layers below the sea bottom. They proposed that the picks, which are considered in our study as $SP$ phases converted on the sea bottom, may rather

represent $PsP$ phases converted at the base of a layer of soft, unconsolidated sediments.

The main reason why these phases cannot be associated with wave conversions in a sediment layer is that in this case, the observed differential $S$-$P$ times would merely depend on the properties of sediments where the station is located and would not be dependent on hypocentral distance. In fact, as we see in Fig. 1a for the observed data, the rays with later $P$-wave times corresponding to larger hypocentral distances always have larger differential times $Ts$-$Tp$.

Another argument that the $PS$ phases were identified correctly was the stable tomographic inversion, and reasonable values of the obtained velocities and $Vp/Vs$ ratio. If we misinterpreted our data and used them in our tomography inversion, we would never obtain such a

good data fit for the S-wave residuals (0.06 s) and such a strong variance reduction (51%) as observed after the inversion in our case. Erroneous data would behave as outliers and would never provide any stable solution. The good fit between the observed and calculated travel times after source locations in the final 3D model is demonstrated in Fig. 1a.

For the study area, we could not directly estimate the properties of the sediment cover and its effect on the seismic wave field, as there were no bottom seismic measurements. However, we can compare this case with the neighboring volcanic area on Gakkel Ridge at 120°E, which is very similar to the 85°E volcano, where ocean bottom seismometers (OBS) were installed[3]. The upper four traces in Fig. 2 represent an example of seismograms recorded by OBS in this area from an event that occurred at a lateral distance of about 8–20 km to the individual stations. The seismograms show very prominent, large-amplitude S-phases on the horizontal channels (red), which cannot be misidentified in the OBS records. Similar S-P travel time differences are seen for the OBS data and for the icefloe data on both volcanoes. At the same time, we cannot identify any notable Ps phases converted at a boundary of the sediments or porous lava layer that should arrive between the P and S phases. It seems unlikely that a Ps phase gets converted to a PsP phase and is visible at seismometers on ice floes, while a much stronger S phase is not converted to a visible SP phase. None of the icefloe seismograms shows later phases that could qualify as potential SP phases following an earlier PsP phase.

To further demonstrate the problem of data processing, Yu and Singh[1] provide the Wadati diagram, in which the estimated $Vp/Vs$ ratio appears to be equal to 1.144, which is too low. We claim that this diagram was constructed inadequately. In Fig. 1b, we show our Wadati diagram constructed for the observed times (red) and modeled times (blue) calculated in the final 3D velocity model. In both cases, we consider the full travel time including a ray segment in the water. In both cases, we see that the dots can be approximated by $(ts_i-ts_j)/(tp_i-tp_j) = 1.71$. To assess how this parameter represents the actual $Vp/Vs$ ratio in solid rocks, we have created two synthetic models based on the 1D velocity distributions with $Vp/Vs$ ratio equal to 2.00 and 1.7. Fig. 1c shows the Wadati diagrams for the data calculated for these two synthetic cases. We obtain the regression coefficients of $(ts_i-ts_j)/(tp_i-tp_j)$ equal to 1.92 and 1.62, respectively, which are lower than the original values. Taking into account this trend, we can estimate that the actual average $Vp/Vs$ ratio for the real case is approximately equal to 1.77, which is a normal value expected in such settings.

Regarding the interpretation issues discussed by Yu and Singh, we should admit that our seismic tomography model alone does not pretend to provide an unambiguous interpretation. The obtained seismic model is only one of the bricks supporting the general concept of volcanism at ultraslow oceanic ridges, which is based on many other elements, such as geochemistry, petrology, geomorphology, numerical modeling etc. In particular, the statement on the volatile-rich magma, besides the tomography model, is based on the direct observations of explosive eruptions and pyroclastic flows at 4 km water depth[4], high $CO_2$ concentration in olivine-hosted melt inclusions (up to ~ 1600 ppm)[5], and anomalous Ba concentrations in bulk-rocks[5]. These data infer that degassing is inevitable at about 13 km depth, which is in good agreement with our seismic tomography images. On the other side, our numerical thermo-mechanical models predict c.a. ten times lower average degree of mantle melting and respectively higher volatile content in extracted melts (as volatile is predominantly partitioned into the melts[6]) at ultra-slow ridges compared to intermediate or fast-spreading ridges.

We should point out that the obtained seismic velocity distributions alone can hardly be converted to petrophysical parameters. In the article, we write: "*we admit that the uncertainties related to the damping definition and the trade-off between the source and velocity parameter determinations do not allow us to provide exact numerical values for the seismic parameters; therefore, we cannot uniquely convert our model into petrological properties*". Therefore, in this study, we interpret our model qualitatively without pretending to provide exact numbers that can be directly converted to melt content, as Yu and Singh[1] propose.

Based on all these arguments, we conclude that our tomographic model is sufficiently trustworthy. Our interpretation and numerical simulations are consistent with all other observations and appear to be reasonable to explain unusual explosive volcanism on the Gakkel Ridge.

## Data availability

The waveform data of the entire experiment are available in Zenodo: Schlindwein, Vera. (2022). Ice-floe-based records of seismic events at 85°E Gakkel Ridge, Arctic Ocean [Data set]. Zenodo. https://doi.org/10.5281/zenodo.7376749.

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

## Author contributions

I.K. and A.J. were responsible for responses related to the tomography inversion. V.S. prepared the responses related to the initial data. M.L., T.G., and A.I. contributed to preparing responses related to numerical modeling, geochemistry, and interpretation.

## Competing interests

The authors declare no competing interests.
