## [Peer Review File · Nature Communications]

REVIEWERS' COMMENTS:

Reviewer #1 (Remarks to the Author):

This is a reply by Dr. Koulakov et al. to a criticizing comment by Dr. Yu and Dr Singh.

As I pointed out in my review comments to the MA manuscript by Dr. Yu and Dr. Singh, the most important discussion here is whether the S-wave velocity model proposed by Dr. Koulakov et al. is accurate or not. This discussion is primarily dependent on the author's assumption that S-P converted phases they used for the tomography are S-waves converted to P-waves on the sea bottom. To rule out other possibilities of conversion patterns and locations, Koulakov et al. argue in their reply that "there is no evidence for a thick sedimentary layer in the study area. On the contrary, the refraction seismic surveys revealed normal oceanic near-surface P-wave velocities (2.4-3.9 km/s)⁶. In addition, gravity coring in the area of the 85°E volcano did not find any trace of soft sediments⁷". After reading the reply, however, I am still concerned that the Vs model has not been properly constructed since the significant delay of S waves is possible even in the absence of sedimentary layers.

It is well known that in an oceanic environment, phase conversions can occur at sharp structural boundaries not only at the seafloor but also below the seafloor. For example, an OBS refraction study close to a spreading center where a thick sedimentary layer is missing (Lata and Dunn, 2020, Marine Geology) indicate that P-to-S converted seismic phases arise from two shallow interfaces, one at ~80 m depth, and the other at 500–650 m depth. They also demonstrate that these shallow layers hold low S-wave velocity (< 1.0 km/s) and high Vp/Vs ratio of over ~5. This means that such a thin layer can produce a remarkable delay of S-wave arrivals as in the study of Dr. Koulakov et al., and thus its effects need to be dealt with adequately in tomography analyses. This is the primary reasons why such tomography studies include "station corrections" for S wave arrivals. However, as far as I understand, Dr. Koulakov et al. has not taken it into account in their calculation.

I also agree with another comment by the criticizing authors that the picked sP waves are actually PsP-waves. However, Dr. Koulakov has not provided any convincing arguments on this.

Based on the considerations above, it seems to me that the authors haven't addressed the main concerns by the criticizing authors.

Reviewer #2 (Remarks to the Author):

This is an important discussion to be published.

(1) whether you judge that the criticism raised in the comment is valid, Yu et al. point out the misinterpretation of the sP phase by Koulakov et al. Looking at Fig. 1S of the original paper and Yu et al, I would agree the point Yu et al. mentioned. In particular small amplitude of Koulakov et al.'s interpreted sP phase does not explain expected waveform of S-to-P converted wave. However, the data presented by Koulakov et al are only the waveform of one earthquake, and I do not have access to other data. I would need to check other data as well to judge the details. I am actually surprised the original paper does not present any other waveforms in the supplement.

With respect to the physical modeling of the Vp/Vs values, the logic of Yu et al. seems reasonable, but both Yu et al. and Koulakov et al. make several assumptions, and there may be different views on the validity of these assumptions.

(2) whether it is likely to be of significant interest to the readers of the original Article.

I believe that it is worth to publish the discussion about the point that there may be misinterpretation of the most important data in the original paper. In addition, the authors of the original paper should disclose the seismic waveform data they used.

(3), could it be stated more concisely?
Yu et al. concisely describes the claim.

REVIEWERS' COMMENTS:

Reviewer #1 (Remarks to the Author):

This is a reply by Dr. Koulakov et al. to a criticizing comment by Dr. Yu and Dr Singh.

As I pointed out in my review comments to the MA manuscript by Dr. Yu and Dr. Singh, the most important discussion here is whether the S-wave velocity model proposed by Dr. Koulakov et al. is accurate or not. This discussion is primarily dependent on the author's assumption that S-P converted phases they used for the tomography are S-waves converted to P-waves on the sea bottom. To rule out other possibilities of conversion patterns and locations, Koulakov et al. argue in their reply that "there is no evidence for a thick sedimentary layer in the study area. On the contrary, the refraction seismic surveys revealed normal oceanic near-surface P-wave velocities (2.4-3.9 km/s)⁶. In addition, gravity coring in the area of the 85°E volcano did not find any trace of soft sediments⁷". After reading the reply, however, I am still concerned that the Vs model has not been properly constructed since the significant delay of S waves is possible even in the absence of sedimentary layers.

REP: We have considerably rewritten the manuscript by expanding the arguments on the validity of data analysis and placing them in the first place in the text.

The main reason why these phases cannot be associated with wave conversions in a sediment layer is that in this case the observed differential *S-P* times would merely depend on the properties of sediments where the station is located and would not be dependent on hypocentral distance. In fact, we clearly see that rays with later *P*-wave times corresponding to larger hypocentral distances always have larger differential times *T_s-T_p*. (L32-36)

Another argument that the *PS* phases were identified correctly was the stable tomographic inversion and reasonable values of the obtained velocities and *V_p/V_s* ratio. If we misinterpreted our data and used them in our tomography inversion, we would never obtain such a good data fit for the *S*-wave residuals (0.06 s) and such a strong variance reduction (51%) as observed after the inversion in our case. Erroneous data would behave as outliers and would never provide any stable solution. (L37-42)

It is not easy to debate about potential layers of sediments in the area considered in the paper in the absence of bottom seismometers and any special studies oriented to such types of shallow structures. Therefore, we have added information about a neighboring volcano zone on Gakkel Ridge having similar geological structure (L43-56). The available bottom seismometer data clearly show very strong *S*-phases, which are expected to be converted to the clear *SP* wave in the water layer. On the other hand, these records did not recognize any prominent *Ps* phase converted in shallow sediments. We can expect that a similar wave configuration exists in the area of our study.

It is well known that in an oceanic environment, phase conversions can occur at sharp structural boundaries not only at the seafloor but also below the seafloor. For example, an OBS refraction study close to a spreading center where a thick sedimentary layer is missing (Lata and Dunn, 2020, Marine Geology) indicate that P-to-S converted seismic phases arise from two shallow interfaces, one at ~80 m depth, and the other at 500–650 m depth. They also demonstrate that these shallow layers hold low S-wave velocity (< 1.0 km/s) and high V_p/V_s ratio of over ~5. This means that such a thin layer can produce a remarkable delay of S-wave arrivals as in the study of Dr. Koulakov et al., and thus its effects need to be dealt with adequately in tomography analyses. This is the primary reasons why such tomography studies include “station corrections” for S wave arrivals. However, as far as I understand, Dr. Koulakov et al. has not taken it into account in their calculation.

REP: Regarding the sediment-converted phases, see our previous response. As for the station corrections, it was not possible to apply them in this study, because every station point while changing locations was considered as another station. It would create too many unknowns and make the inversion unstable.

I also agree with another comment by the criticizing authors that the picked sP waves are actually PsP-waves. However, Dr. Koulakov has not provided any convincing arguments on this.

Based on the considerations above, it seems to me that the authors haven't addressed the main concerns by the criticizing authors.

REP: In the new version of the manuscript, we have provided several additional arguments proving that the picked SP phases are real and not misinterpreted with PsP phases generated by conversions in the sediment layer (see our comments above).

Reviewer #2 (Remarks to the Author):

This is an important discussion to be published.

(1) whether you judge that the criticism raised in the comment is valid, Yu et al. point out the misinterpretation of the sP phase by Koulakov et al. Looking at Fig. 1S of the original paper and Yu et al, I would agree the point Yu et al. mentioned. In **particular small** amplitude of Koulakov et al.'s interpreted sP phase does not explain expected waveform of S-to-P converted wave.

REP: We do not understand this comment. In our opinion it is not easy to predict in this setting the amplitudes of a SP converted phase given that the seismometers are installed on a drifting ice floe. We base our phase identification on the knowledge that S phases in ocean bottom records in these settings are very strong and likely powerful enough to produce SP converted energy at the sea surface while converted phases from shallow interfaces have very small amplitudes in comparison to the S phase.

However, the data presented by Koulakov et al are only the waveform of one earthquake, and I do not have access to other data. I would need to check other data as well to judge the details. I am actually surprised the original paper does not present any other waveforms in the supplement.

REP: We have provided the access to the full waveform database at Zenodo: Schlindwein, Vera. (2022). Ice-floe based records of seismic events at 85°E Gakkel Ridge, Arctic Ocean [Data set]. Zenodo. <https://doi.org/10.5281/zenodo.7376749>

With respect to the physical modeling of the V_p/V_s values, the logic of Yu et al. seems reasonable, but both Yu et al. and Koulakov et al. make several assumptions, and there may be different views on the validity of these assumptions.

REP: In our previous version of the response, we have demonstrated that Yu and Singh used a wrong way of calculation of the Wadati diagram and therefore they obtained unrealistic values of V_p/V_s ratio. We have provided the calculations for both observed and synthetic data and estimated that the V_p/V_s ratio is equal to 1.77, which is a reasonable value for these settings.

(2) whether it is likely to be of significant interest to the readers of the original Article. I believe that it is worth to publish the discussion about the point that there may be misinterpretation of the most important data in the original paper. In addition, the authors of the original paper should disclose the seismic waveform data they used.

REP: Based on several arguments presented in the response file, we claim that any misinterpretation of our initial data is very unlikely. As we said earlier, the waveforms are now available in open access.

(3), could it be stated more concisely?
Yu et al. concisely describes the claim.

Referees' comments:

Reviewer #1 (Remarks to the Author):

This is the second review on the reply by Dr. Koulakov et al. to the MA by Dr. Yu and Dr Singh.

In the rebuttal letter and the revised manuscript, the authors added new information on Ts-Tp that "In fact, we clearly see that rays with later P-wave times corresponding to larger hypocentral distances always have larger differential times Ts-Tp. (L32-36)".

If this is true, it can be supportive for their phase identifications.

However, it is just described in a qualitative manner in the text.

I suggest adding some figures demonstrating it in a quantitative way, such as waveforms data with P- and S-wave picks (or Ts-Tp itself) aligned as a function of hypocentral distance.

If such figures show the clear proportional relationship between them,

it can prove that S-wave time delay occurs on the whole ray paths and that sedimentary effects are minimal,

and then readers will be fully convinced.

Reviewer #2 (Remarks to the Author):

I do not believe the authors have adequately responded to reviewer 1's first comment. Reviewer 1 cites an example of a conversion at an interface at 500 m depth in the crust (probably within the igneous crust) covered with the very thin sediment, but the authors only discuss "sediment-converted phases" in their reply, but the authors do not consider the conversion interface in the crust pointed out by reviewer 1.

The comment exchanged by Koulakov et al. and Yu and Shigh discusses the validity of assumptions made when analyzing data obtained in special environments, and I believe that the readers of Nature Communication (and the editors) should judge their arguments. From that point of view, the exchanged comment of both authors deserves publication in Nature Communication.

Rebuttal letter on the manuscript “Reply to: Incorrect evidence for the presence of melt at the explosive volcano at the Gakkel Ridge” by Ivan Koulakov, Vera Schlindwein, Mingqi Liu, Taras Gerya, Andrey Jakovlev, and Aleksey Ivanov

(the authors' responses are highlighted with red and indicated with “REP”)

Reviewer #1 (Remarks to the Author):

This is the second review on the reply by Dr. Koulakov et al. to the MA by Dr. Yu and Dr Singh.

In the rebuttal letter and the revised manuscript, the authors added new information on Ts-Tp that "In fact, we clearly see that rays with later P-wave times corresponding to larger hypocentral distances always have larger differential times Ts-Tp. (L32-36)". If this is true, it can be supportive for their phase identifications. However, it is just described in a qualitative manner in the text. I suggest adding some figures demonstrating it in a quantitative way, such as waveforms data with P- and S-wave picks (or Ts-Tp itself) aligned as a function of hypocentral distance. If such figures show the clear proportional relationship between them, it can prove that S-wave time delay occurs on the whole ray paths and that sedimentary effects are minimal, and then readers will be fully convinced.

REP: We have included Figure 1a with the distributions of the observed and modeled P and S wave travel times after source locations in the final velocity model versus hypocentral distance (between the source point and station on the sea surface). In this figure, we clearly observe that the observed travel times form two distinct branches with increased S-P times for larger distances. The calculated travel times match the observed times rather well, which demonstrates the adequacy of the final velocity model and the correctness of the phase identification.

The other figures have been re-arranged correspondingly.

Reviewer #2 (Remarks to the Author):

I do not believe the authors have adequately responded to reviewer 1's first comment. Reviewer 1 cites an example of a conversion at an interface at 500 m depth in the crust (probably within the igneous crust) covered with the very thin sediment, but the authors only discuss "sediment-converted phases" in their reply, but the authors do not consider the conversion interface in the crust pointed out by reviewer 1.

REP: We do not understand this comment. The reviewer claims that the 1st reviewer mentions “igneous crust” whereas we discuss the effect of wave conversion in a sedimentary layer. Actually, we cannot find any mentioning of “igneous crust” in the comments of the 1st reviewer. When saying about possible phase misinterpretations, he always mention “the sedimentary layer”. For example, in the last comment, there is a phrase “sedimentary effect” (highlighted) and no other causes of wave conversion. We do not see what we can correct according to this comment.

The comment exchanged by Koulakov et al. and Yu and Shigh discusses the validity of assumptions made when analyzing data obtained in special environments, and I believe that the readers of Nature Communication (and the editors) should judge their arguments. From that point of view, the exchanged comment of both authors deserves publication in Nature Communication.